photochemistry/materials science/ nanotechnology

Bi$_2$S$_3$ thin films, pulse-plating, nano-bismuth, bismuth potassium citrate, photoelectrochemical

**Author for correspondence:**
Guochen Zhao
e-mail: zhaogch@sdas.org

# Synthesis of Bi$_2$S$_3$ thin films based on pulse-plating bismuth nanocrystallines and its photoelectrochemical properties

Fangchang Ding[1], Qiujin Wang[1], Shaofei Zhou[1], Guochen Zhao[2], Ying Ye[1] and Reza Ghomashchi[3]

[1]Institute of Marine Geology and Resources, Ocean College, Zhejiang University, Zhoushan 316021, People's Republic of China
[2]Shandong Provincial Key Laboratory of High Strength Lightweight Metallic Materials, Advanced Materials Institute, Qilu University of Technology (Shandong Academy of Sciences), Jinan 250000, People's Republic of China
[3]School of Mechanical Engineering, The University of Adelaide, Adelaide, South Australia 5005, Australia

QW, 0000-0001-9961-3533; GZ, 0000-0002-2493-6343; RG, 0000-0003-3633-2296

The solubility of Bi$^{3+}$ in aqueous solution is an important factor that limits the fabrication of high-quality Bi$_2$S$_3$ thin films. In order to find a low-cost method to manufacture high-quality Bi$_2$S$_3$ thin films, we are reporting the preparation of the Bi$_2$S$_3$ thin films based on pulse-plating method in this paper for the first time. The nano-bismuth particles were obtained by electroplating on fluorine-doped SnO$_2$ (FTO)-coated conducting glass substrates with saturated bismuth potassium citrate solution as the electroplating bath, and then it was put into a muffle furnace to oxidize. Finally, the thin films depositing on FTO glass substrates were put into the thioacetamide solution for vulcanization. In the end, the Bi$_2$S$_3$ thin films were successfully prepared on FTO glass substrates. Different characterization techniques were used to characterize the structure, morphology and photoelectrochemical properties of the prepared thin films. The test results revealed that we used this method to synthesize the high-quality Bi$_2$S$_3$ thin films, thus the Bi$_2$S$_3$ materials synthesized through this method are promising candidates in photoelectrochemical application.

# 1. Introduction

Bismuth sulfide (Bi$_2$S$_3$) is an important inorganic semiconductor material of the V–VI group. It has a suitable band gap

($E_g = 1.3$–1.7 eV) [1], which is very close to the optimal absorption energy band gap of solar cells [2]. The $Bi_2S_3$ semiconductor can extend the absorption width of sunlight to the near-infrared band and have a high absorption coefficient [3]. Moreover, it has a relatively high carrier mobility [4]. It is rapidly finding its way into the forefront of advanced materials due to its stable, non-toxic and environmentally friendly nature and superior electronic, magnetic, optical, catalytic and mechanical properties. One of its potential applications is in photovoltaic devices [5,6]. This includes solar cells [7], Schottky diodes [8], sensors [9], thermoelectric devices [10] and photodetectors [11]. The potential applications have even expanded after discovering that bismuth sulfide may be synthesized as nanorods [12], nanotubes [13] and nanowires [14].

$Bi_2S_3$ is considered as a good electrode material for solar cells due to its good photoconductivity and useful photoelectric properties and has been widely used in the field of photo-electrochemistry [15]. Using high-aspect-ratio nano- and micro-structured semiconductors with radial p-n junctions is an important strategy to design efficient solar cells [16]. Considering its high surface area and low light reflectivity, the prospect of nanostructured $Bi_2S_3$ thin film is promising for using solar energy. Therefore, the fabrication of nano-sized bismuth sulfide with different morphologies may improve its various properties and performances as a semiconducting material in devices [17].

For the preparation and characterization of $Bi_2S_3$ thin films, new technology is constantly developing. In recent years, the methods commonly used to synthesize bismuth sulfide thin films include electrodeposition [18], rapid thermal evaporation [4], chemical bath deposition [19] and metal-organic chemical vapour deposition [20]. The properties of the thin films fabricated by the above methods are different, depending on their chemical composition and crystal structure. Recently, Lin & Lee [21] prepared bismuth sulfide semiconductor-sensitized tin dioxide solar cells by successive ionic layer adsorption and reaction (SILAR) process at room temperature. This is a new type of solar cell construction by coating bismuth sulfide nanoparticles on tin dioxide electrodes to produce liquid-connected solar cells. The photoelectrochemical properties of the prepared $Bi_2S_3$ thin films by this method were characterized by other researchers using a xenon lamp as the light source [20]. A photocurrent density of 1 mA cm$^{-2}$ was reported under the light intensity of 40 mW cm$^{-2}$ [22]. The value of photocurrent density is among the highest reported for any $Bi_2S_3$ photoelectrode to date.

The SILAR technique for the preparation of $Bi_2S_3$ thin films from aqueous solution is a simple and economic technique [23]. However, the solubility of $Bi^{3+}$ in aqueous solution is very low which limits the preparation of the high-quality $Bi_2S_3$ thin films. In order to resolve this problem, among the methods mentioned above, the solvothermal process [24] is the most popular technology for preparing $Bi_2S_3$ thin films based on a chemical process. Although there has been significant progress for solvothermal technology in the preparation of bismuth sulfide thin films so far, the current state of the art suffers a range of drawbacks such as the need for relatively high temperature and pressure, long reaction time and application of toxic organic solvents [25].

To circumvent the above-mentioned issues in the preparation of bismuth sulfide thin films and be consistent with the concept of green chemistry, we have employed a simple and economical green synthesis method to fabricate $Bi_2S_3$ thin films. This is based on electroplating as a conventional effective technology to electrochemically deposit metal or alloy films in solutions. It has numerous advantages including fast deposition rate, adjustable film thickness and selective deposition assisted by pre-prepared patterns conducted at room temperature and atmospheric pressure [26]. In this method, the non-toxic and harmless saturated bismuth potassium citrate solution is used as an effective electroplating solution that is easily soluble in aqueous solution. Compared with the electroplating method for the preparation of $Bi_2S_3$ thin films, we do not use the toxic non-aqueous dimethyl sulfoxide medium, the complexing agent and the surfactant [27].

Compared with previous studies, our reported method is able to deposit more uniform $Bi_2S_3$ thin films with higher photocurrent density and excellent photoelectric conversion properties. In addition, the preparation method is relatively simple and low cost with potential mass production of $Bi_2S_3$ thin films at industrial scale.

# 2. Experimental procedures

## 2.1. Materials

Thioacetamide (TAA), anhydrous ethanol, acetone, nitric acid and bismuth potassium citrate ($C_{12}H_{10}BiK_3O_{14}$) were purchased commercially. All reagents were of analytical grade (purity greater

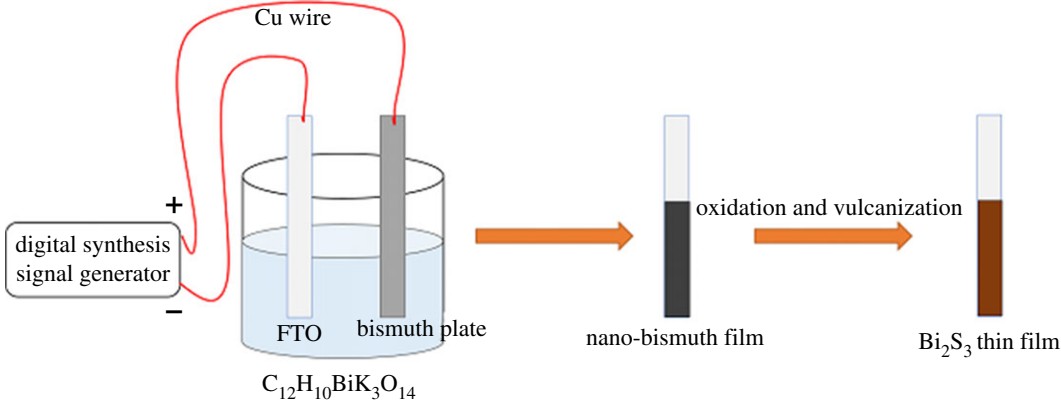

**Figure 1.** Schematic representation of the synthesis of $Bi_2S_3$ by pulse-plating method.

than 97%) and used in the as-received condition without any further purification treatment. Deionized water was used throughout this experiment.

## 2.2. Bismuth sulfide thin film preparation

First, a saturated bismuth potassium citrate solution was prepared and stored in 250 ml volumetric flask at room temperature. The fluorine-tin-oxide (FTO) glass substrates (2 × 4 cm) were ultrasonically cleaned in acetone, anhydrous ethanol and deionized water for 20 min in each solution, respectively, and dried under nitrogen and stored in a desiccator. The first step in the synthesis of $Bi_2S_3$ thin film is the deposition of a nano-bismuth layer. The nano-bismuth layer was obtained by the method of sinusoidal pulse-plating. The cleaned FTO glass substrate was connected to the cathode of the SG1020A digital synthesis signal generator. Then, the anode of the SG1020A digital synthetic signal generator was connected to the metal bismuth plate (2 × 3 cm, purity greater than 99.99%) through Schottky diode, so that the anode can obtain the positive half axis of the sinusoidal pulse. The amplitude, the frequency and the direct current (AC) offset voltage of the sinusoidal pulse were 10 V, 600 Hz and 5 V, respectively. The whole electroplating process was carried out at room temperature for preparing nano-bismuth. After electroplating, the FTO glass substrate was washed in deionized water for 20 s and then left to dry naturally in air.

The nano-bismuth coating on the FTO glass substrate was put into a muffle furnace with a heating rate of 5°C min$^{-1}$. The samples were oxidized in the muffle furnace at 300°C, 400°C and 500°C for 4 h, respectively, and cooled naturally in the switched-off muffle furnace. The whole process was carried out in the atmospheric environment without any gas flow. The oxide films were submerged in 0.1 M thioacetamide solution heated to 80°C in a digital thermostat-controlled water bath. The samples were kept for 10 min to vulcanize and form the $Bi_2S_3$ compound. The samples were then taken out and washed with deionized water and dried naturally. The schematic of the synthesis of $Bi_2S_3$ by pulse-plating method is illustrated in figure 1.

## 2.3. Characterization

The chemical compositions of the prepared thin films were checked and analysed by X-ray diffraction (XRD, X'Pert3 Powder) using Cu K$_\alpha$ radiation ($\lambda = 0.15416$ Å) source. The X-ray machine was operated at 40 kV, 40 mA within the 2$\theta$ range of angles between 20° and 80°. The surface topography, nanoparticle size and microstructure of the thin films were characterized using a scanning electron microscope (SEM, Sigma 500). The optical absorption properties and the band gap of the $Bi_2S_3$ thin films were measured in diffuse reflectance mode on an UV-Vis spectrophotometer (OPTIZEN 3220UV, Mecasys) with an integrating-sphere accessory. All photoelectrochemical measurements of the prepared thin films were carried out using the Chenhua CHI660E electrochemical workstation (Shanghai, China) under a three-electrode system. The prepared $Bi_2S_3$ thin films, the Ag/AgCl electrode and the platinum electrode were used as working electrodes, reference electrode and counter electrode, respectively. A 0.5 M $Na_2SO_4$ solution served as the electrolyte solution for photoelectrochemical measurements, which were all de-aerated by nitrogen for 20 min before experiments. The electrochemical impedance spectroscopy was performed

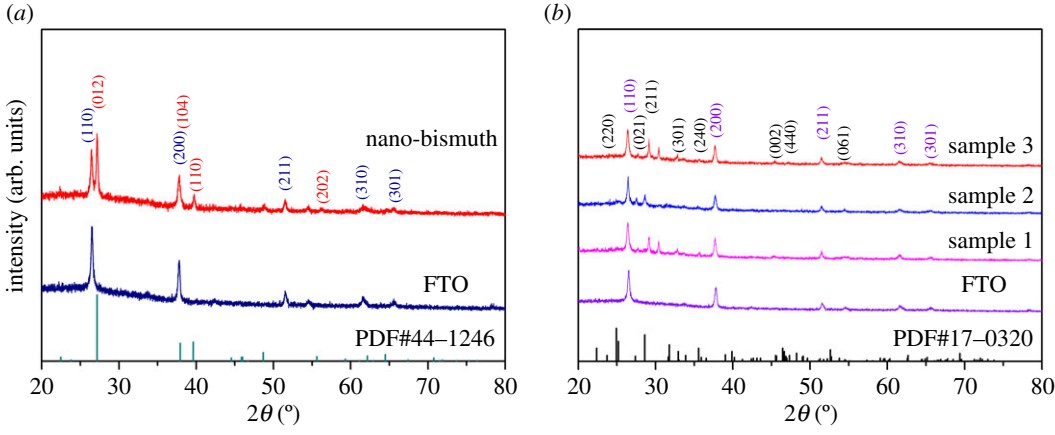

**Figure 2.** The XRD diffraction patterns of the thin films prepared on FTO substrate. (*a*) nano-bismuth coating. (*b*) $Bi_2S_3$ films, prepared at 300℃ (sample 1), 400℃ (sample 2) and 500℃ (sample 3).

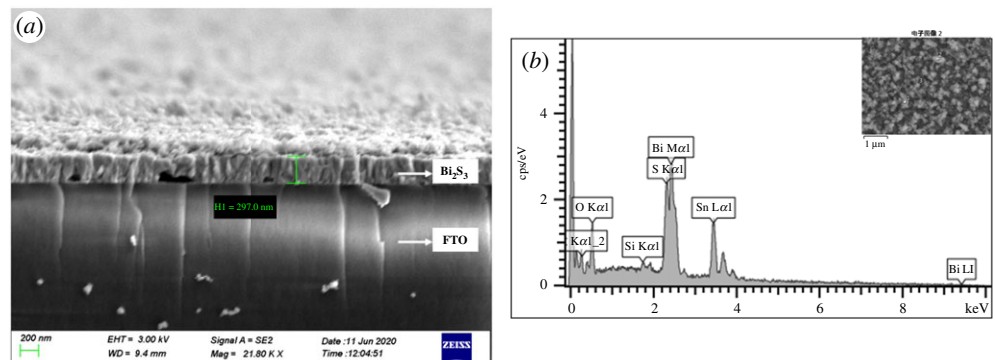

**Figure 3.** (*a*) SEM image and (*b*) EDS pattern of $Bi_2S_3$ thin film structures (sample 1 (300℃)).

**Table 1.** The size of the bismuth nanocrystalline.

| (hkl) | $2\theta$ (deg) | FWHM (deg) | D (nm) |
| --- | --- | --- | --- |
| (012) | 27.170 | 0.225 | 35.920 |
| (110) | 39.697 | 0.348 | 24.000 |
| (202) | 48.897 | 0.506 | 17.054 |

by applying an AC voltage of 10 mV versus Ag/AgCl with the frequency range from 0.1 Hz to 100 kHz under visible light illumination.

# 3. Results and discussion

The X-ray diffraction (XRD) spectra of the thin films are shown in figure 2. The thickness of the $Bi_2S_3$ film is about 297 nm according to the SEM image (figure 3*a*) and the size of the bismuth nanocrystalline is about 25 nm according to the XRD data by Scherrer method (see table 1). The XRD peaks in figure 2*a* are from nano-bismuth coating that is compared with the bismuth standard pattern. From top to bottom is the XRD diffraction pattern of prepared nano-bismuth on FTO glass substrate, the XRD diffraction pattern of FTO glass and the standard diffraction pattern of bismuth metal (JCPDS: PDF number 44–1246). It is clear that the diffraction peaks of the prepared nano-bismuth samples perfectly match with the standard diffraction data of bismuth, and no diffraction peaks of any impurities are found, indicating the successful preparation of a relatively pure nano-bismuth thin film. Figure 2*b* is the XRD pattern of the successfully prepared $Bi_2S_3$ thin film samples, the diffraction peaks of FTO

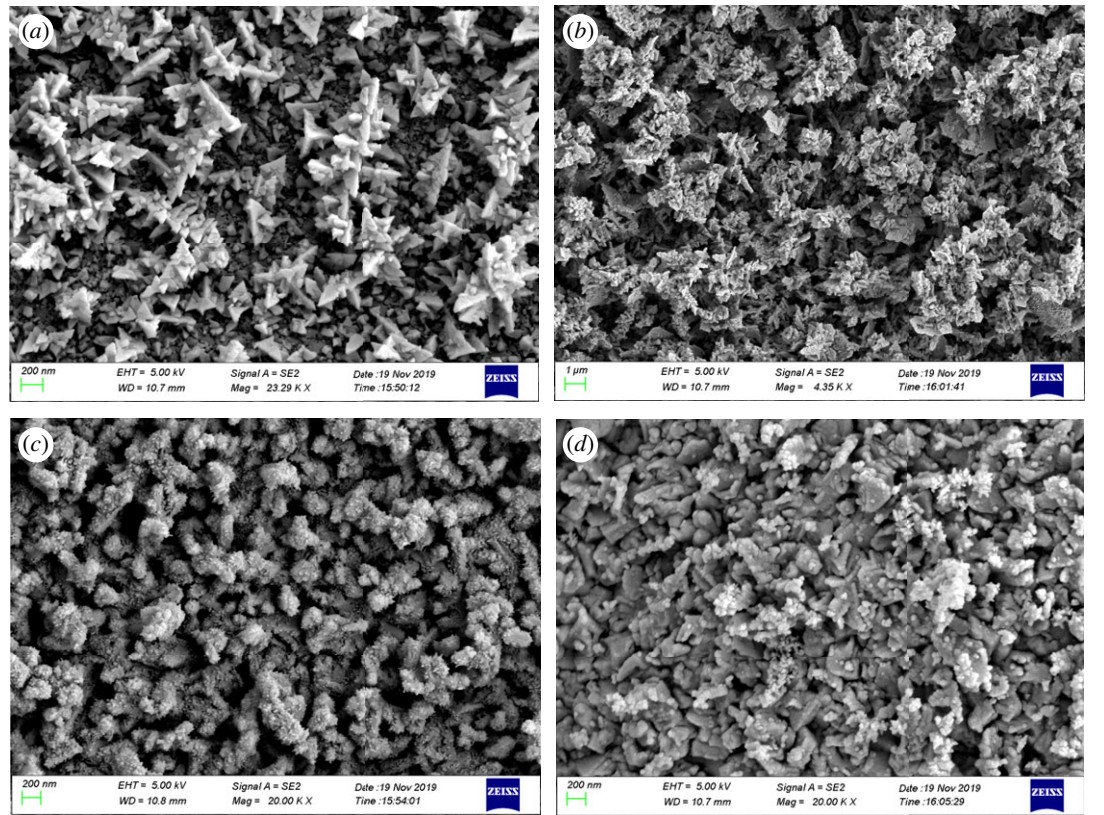

**Figure 4.** SEM images of thin films preparation on FTO glass. (*a*) Nano-bismuth preparation by pulse-plating. Morphology of $Bi_2S_3$ thin films prepared under different temperatures (*b*) 300℃ (sample 1), (*c*) 400℃ (sample 2), (*d*) 500℃ (sample 3).

glass substrate and the standard diffraction pattern of bismuth sulfide (JCPDS: PDF number 17–0320), the structure of $Bi_2S_3$ belongs to the orthorhombic system, space group Pbnm. The intensities of bismuth sulfide peaks are relatively weak due to the influence of amorphous nature and strong background peaks of the FTO glass substrate. Among the samples, there are no diffraction peaks of bismuth oxide (JCPDS: PDF number 74–1374). Since the thickness of $Bi_2S_3$ film is only around 300 nm, the diffraction is weak. The X-rays can easily penetrate and reach the interface of FTO substrate. In order to figure out the composition of film, EDS analysis was carried out, which demonstrated the existence of elements Bi and S (the atomic ratio is around 2 : 3) in figure 3*b*. The other element peaks belong to the FTO substrate. Thus, it can be concluded that the bismuth oxides are mostly converted to $Bi_2S_3$. The XRD patterns in figure 2*b* confirm that the application of different temperatures for the coating preparation did not change the structural composition of the FTO glass substrate.

The surface morphology of the thin films deposited on the FTO glass substrates is shown in figure 4. Figure 4*a* reveals the morphology of the nano-bismuth layer comprising nano-bismuth grains of about 100 nm. The typical SEM images of $Bi_2S_3$ thin films under different treatment conditions are shown in figure 4*b*–*d*. It can be clearly seen that the grains of bismuth sulfide thin films under different preparation conditions are all at the nanometre size. Figure 4*b* shows the morphology of $Bi_2S_3$ thin film prepared by oxidizing the nano-bismuth at 300℃ in the muffle furnace. The thin film appears to consist of grains formed from sintered and agglomeration fine nanoparticles.

For the $Bi_2S_3$ thin film fabricated at 400℃, figure 4*c*, the surface of the sintered agglomerated grains are somewhat smoothened off and composed of pillars formed by bismuth sulfide nanoparticles with smaller grains. The effect of higher temperature in encouraging greater diffusion is clear here when compared with specimens oxidized at 300℃. The effect of even higher oxidizing temperature of 500℃ is detected in figure 4*d*, where the morphology of the $Bi_2S_3$ thin film is no longer as spongy as those of 300 and 400℃ samples. The morphology of the prepared $Bi_2S_3$ thin films is such that it creates a fully fused large surface area with a tight contact structure. Compared with nanorods structure [25], this structure facilitates the migration and transfer of the electrons.

UV-Vis spectroscopy was also performed to probe the energy band structures in the range of 400 to 1000 nm at room temperature. Figure 5*a* shows the representative UV-Vis absorption spectrum of $Bi_2S_3$

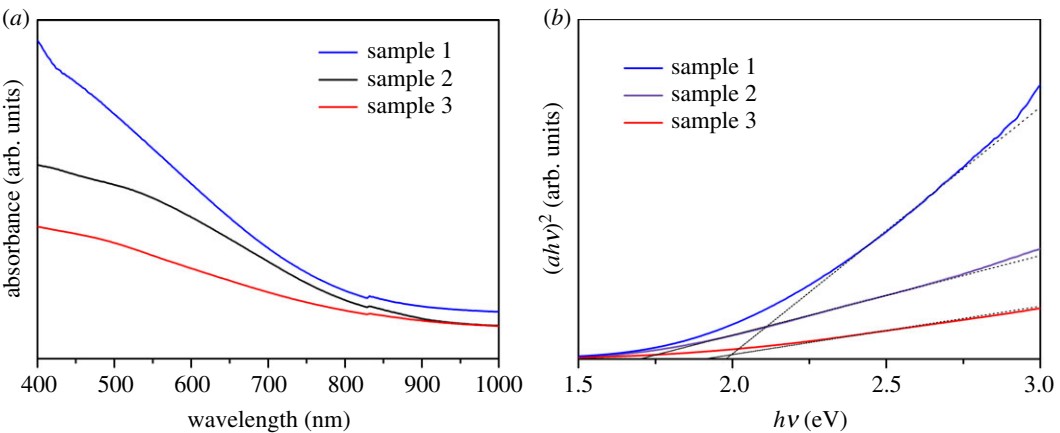

**Figure 5.** (a) UV-Vis absorption spectrum of Bi$_2$S$_3$ thin film samples (sample 1 (300℃), sample 2 (400℃), sample 3 (500℃)) and (b) Tauc plot of $(\alpha h\nu)^2$ versus ($h\nu$) for the value of band gap estimation.

thin film samples under different prepared conditions. It can be observed that Bi$_2$S$_3$ thin film samples showed a good correspondingly wide photoabsorption range in the visible light region. The curves in figure 5a reveal that the light absorption intensity of sample 1 is the largest under different treatment conditions, indicating that the surface topography of sample 1 is beneficial to improve the photoelectron capture efficiency. This may be due to the nature of the surface topography of sample 1 which reflects the incident beam in many directions causing large dispersion with a limited amount of reflected light reaching the detector. In addition, it is known that the optical absorption properties of semiconductor materials are closely related to their optical energy band gap. The corresponding band gap values of the prepared thin film samples were determined by the Tauc relationship and the properties of the direct-transition for bismuth sulfide [22]. It can be seen from figure 5b that the band gap values of the thin film samples are around 1.7 eV. The band gap values for sample 1, sample 2 and sample 3 are about 1.68, 1.93 and 1.95 eV, respectively, which are similar to previous reports [28]. From the electrochemical AC impedance spectra (EIS), it can be seen that the prepared Bi$_2$S$_3$ thin films have fast carrier transfer, and high electron–hole separation efficiency. These results are favourable for the samples to produce a large photocurrent.

In order to clarify the separation and transportation of photo-generated carriers for the Bi$_2$S$_3$ thin films, the Chenhua CHI660E electrochemical workstation was employed to measure the photoelectrochemical characteristics of the as-fabricated Bi$_2$S$_3$ films in a three-electrode configuration. The Bi$_2$S$_3$ thin films prepared on the FTO glass substrate (the effective photosensitive area is 2 × 2 cm), the Ag/AgCl electrode and the platinum electrode acted as the working, reference and counter electrode, respectively. The photoelectrochemical properties of the Bi$_2$S$_3$ thin films were measured in 0.5 M Na$_2$SO$_4$ electrolyte. A xenon lamp was used as the experiment light source with the light intensity kept at 100 mW cm$^{-2}$. The photovoltage–time, photocurrent density–time and electrochemical impedance spectroscopy were characterized.

The corresponding photocurrent density–time and photovoltage–time curves of the Bi$_2$S$_3$ thin films are shown in figure 6. The test process is measured at room temperature. The photocurrent density–time curves were obtained under the condition of 0 V bias in figure 6a. It can be observed from figure 6a that the photocurrent density curves of all the samples remained stable under the conditions of light and darkness. The generated photocurrent density of the Bi$_2$S$_3$ thin films is about 25 mA cm$^{-2}$. When the light source irradiated on the working electrode, all the samples displayed a significant photoresponse and rapidly generated anode current. After the light source is turned off, the photocurrent of the working electrode quickly disappears. The measurement results of the photocurrent demonstrated that the prepared Bi$_2$S$_3$ thin film samples occupy excellent electron–hole separation and transportation. The generated photocurrent density of the Bi$_2$S$_3$ thin films prepared by us is much higher than that generated of the Bi$_2$S$_3$ nanorods prepared by previous work [25], which may be attributed to it having a large surface area with a tight contact structure. The increase of surface roughness will enhance the reflection effect of incident light on the interface and thus improve the photo-conversion efficiency. Figure 6b is the open-circuit photovoltage of the Bi$_2$S$_3$ thin film sample, as it can be observed, after stopping the illumination the photovoltage shows a slow

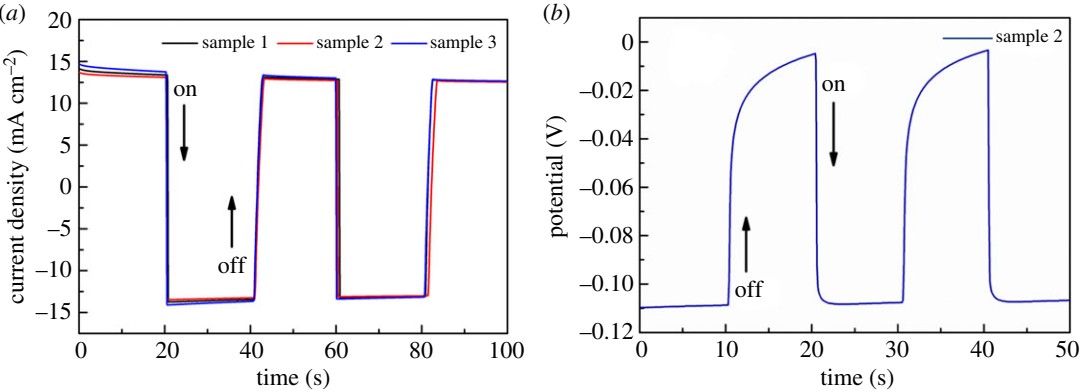

**Figure 6.** (a) Photocurrent density–time plot of $Bi_2S_3$ thin films samples (sample 1 (300℃), sample 2 (400℃), sample 3 (500℃)) and (b) photovoltage–time plot of the $Bi_2S_3$ thin films (sample 2 (400℃)).

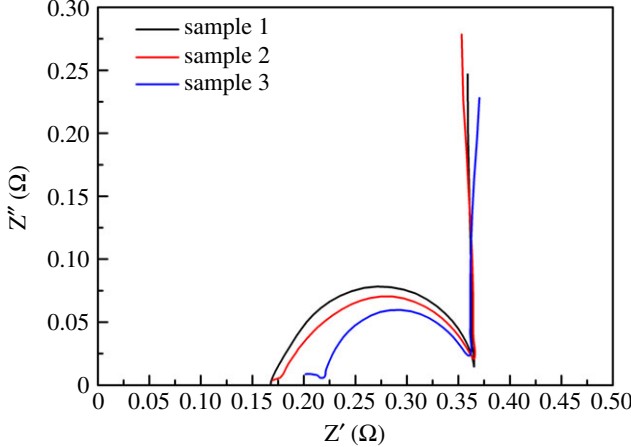

**Figure 7.** Nyquist plots of $Bi_2S_3$ thin films samples (sample 1 (300℃), sample 2 (400℃), sample 3 (500℃)) under illumination and open-circuit potential conditions.

attenuation. Under the open-circuit conditions, electrons are collected in the semiconductor nanostructure film under the irradiation of visible light, and the Fermi level is shifted to a negative potential. Once the light source is stopped, the accumulated electrons are slowly released. This slow decay indicates that the excited electrons can survive longer, so it can promote electron transport without loss at the grain boundaries [29]. The results show that the materials of the prepared $Bi_2S_3$ films have better stability.

EIS can be used to study the electron mobility in the photo-anode film and the electron exchange capability between the photo-anode and the electrolyte interface [30]. This technique was used to further characterize the electron–hole separation efficiency of the prepared $Bi_2S_3$ thin film samples. Figure 7 is the AC impedance spectrum curves of the prepared $Bi_2S_3$ thin film samples as the photo-anode in a 0.5 M $Na_2SO_4$ solution under illumination and open-circuit potential conditions. It can be seen from the results of the Nyquist curves that the impedance of the $Bi_2S_3$ thin films as a photo-anode is quite small. This also shows that the materials of the prepared $Bi_2S_3$ thin films as photo-anodes can significantly promote the effective separation of photo-generated electrons and holes. The Nyquist curve in figure 7 consists of arcs and straight lines, indicating that the electrode process is controlled by the charge transfer process and the diffusion process [31]. The reasons for the deviation of the diffusion impedance straight line may be attributed to the electrode surface roughness [32]. On the other hand, in addition to the electrode potential, there are other state variables that cause changes during the measurement. Furthermore, the slope coefficient of the straight line is relative to the ion transfer rate. Thus, the lower value of the charge transfer resistance and the more vertical straight line are beneficial for enhancing the ion transport and thus improving the electrochemical performance [33]. From the basis of the above results, it could be concluded that the materials of the prepared $Bi_2S_3$ thin films have strong photo-adsorption, fast carrier transfer, and high electron–hole separation efficiency.

# 4. Conclusion

In summary, a novel and economical fabrication method was proposed for nano-$Bi_2S_3$ thin film. A saturated bismuth potassium citrate solution was used as the electroplating solution to deposit a nano-bismuth coating on FTO glass substrate by pulse-plating at room temperature. The nano-bismuth film on FTO substrate was then converted successfully to the $Bi_2S_3$ thin films through vulcanization. The UV-Vis absorption spectroscopy showed that the $Bi_2S_3$ thin films prepared by the proposed novel method have a better photoabsorption range in the visible region. In addition, the results of the photoelectrochemical properties measurement of the $Bi_2S_3$ thin films indicated an improved photocurrent density of about $25\,mA\,cm^{-2}$ under the light intensity of $100\,mW\,cm^{-2}$. The photocurrent density is among the highest values reported for any $Bi_2S_3$ photoelectrode to date. The results of EIS have also shown that the fabricated $Bi_2S_3$ thin films as the photo-anode can significantly promote the effective separation of electrons and holes.

The prepared $Bi_2S_3$ semiconductor material has the appropriate optical band gap and excellent photoelectrochemical properties, thus it has a wide application prospect in the field of photoelectrocatalysis. Moreover, the proposed method is simple and economical to synthesize high-quality, high-photosensitivity and low-cost $Bi_2S_3$ thin films.

Ethics. This article does not present research with ethical considerations.

Data accessibility. The data are available at the Dryad Digital Repository: doi:10.5061/dryad.zpc866t63 [34].

Authors' contributions. F.D. and G.Z. carried out the laboratory work, participated in data analysis, participated in the design of the study and drafted the manuscript; R.G. critically revised the manuscript; Y.Y. conceived of the study, designed the study and coordinated the study. S.Z. and Q.W. collected field data and carried out the data analyses. All authors gave final approval for publication and agree to be held accountable for the work performed therein.

Competing interests. There are no conflicts of interest declare.

Funding. This work was supported by the Shandong Province Key Research and Development Plan (2019GGX102047) and the Several Policies on Promoting Collaborative Innovation and Industrialization of Achievements in Universities and Research Institutes (2019GXRC030).

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
