## [Reviewer comments · Royal Society Open Science]

Review History

RSOS-200479.R0 (Original submission)

Review form: Reviewer 1

Is the manuscript scientifically sound in its present form?

No

Are the interpretations and conclusions justified by the results?

No

Is the language acceptable?

No

Do you have any ethical concerns with this paper?

No

Have you any concerns about statistical analyses in this paper?

Yes

Recommendation?

Major revision is needed (please make suggestions in comments)

Comments to the Author(s)

This present work described a pulse-plating method for the synthesis of Bi₂S₃ thin films and also investigated the photoelectrochemical performance. In the present state, the manuscript is badly organized and there are still many problems to be addressed before acceptance for publication.

1. For the synthesis method, the authors claimed that the Bi₂S₃ thin films were prepared based on pulse-plating method in this paper for the first time. So, what is the remarkable progress compared to the electroplating method for the preparation of Bi₂S₃ thin films (J Solid State Electrochem 2009, 13, 1339–1350, 10.1007/s10008-008-0679-z)?

2. The Bi₂S₃ thin films on FTO surface were prepared primarily from the deposition of Bi films by using electroplating method, followed by thermal treatment for oxidation at different temperature and finally converted to Bi₂S₃ films. How can the bismuth oxides be guaranteed totally to be converted to Bi₂S₃ at 80 C for 10 min? From the evidence of PXRD patterns, the diffraction peaks of bismuth-metal films could be remarkably recorded being indexed to JCPDS: PDF number 44-1246. However, the characteristic peaks of Bi₂S₃ films as ascribed to JCPDS: PDF number 17-0320 were pretty weak, and they didn't provide other firm characterization for the chemical structure. How did the authors confirm the formation of Bi₂S₃ films deriving from the precursor oxides?

3. More experimental details should be provided. For examples, how were the UV-Vis adsorption spectra recorded? In transmission mode, reflection mode or other mode? What were the setup parameters for PXRD pattern performing? The testing conditions for photoelectrochemical measurements should be clearly described such as the purification of electrolyte solution by inert gas. The EIS tests should be clarified by providing the stable open-circuit potential value and frequency range.

4. The authors described that the reported method in the present work is able to deposit more uniform Bi₂S₃ thin films. However, from the SEM images, the roughness of Bi₂S₃ films seems very large. Could the authors compare the prepared Bi₂S₃ here to the previous works? Does the surface topography have the effect on the photoelectrochemical performance of Bi₂S₃ films?

5. This work showed the photoelectrochemical performance of as-synthesized Bi₂S₃ films. As demonstrated in literatures, the preferentially growing orientation of Bi₂S₃ nanorod along with [001] direction provides a rapid electron transport route for the photo-induced charge carriers (Dalton Trans., 2012, 41, 5581-5586; Crystengcomm. 2019, 21, 1474-1481). Do the authors could distinguish the orientation of Bi₂S₃ in the films? What do dominate the good photo-conversion efficiency of Bi₂S₃ films in the present work? How was the stability of Bi₂S₃ films, which is usually shown in the manuscript or electronic supplementary materials? How about the electron-hole separation efficiency compared to Bi₂S₃-based composite films with other semiconductors? The comparison of photoelectrochemical performance of as-prepared Bi₂S₃ films to the previous ones should also be discussed.

6. For the Nyquist plot, why did the linear part corresponding to the diffusion process keep in perpendicular to Z' axle rather than the line with a slop of around 1?

7. The language used in the manuscript should be largely polished as any grammar mistakes were found. Such as, "Nanostructured Bi₂S₃ thin film with high surface area and low light reflectivity is one of the most promising solar cells."

Review form: Reviewer 2

Is the manuscript scientifically sound in its present form?

No

Are the interpretations and conclusions justified by the results?

Yes

Is the language acceptable?

Yes

Do you have any ethical concerns with this paper?

No

Have you any concerns about statistical analyses in this paper?

No

Recommendation?

Major revision is needed (please make suggestions in comments)

Comments to the Author(s)

The authors report the synthesis and photoelectrochemical properties of Bi₂S₃ thin films samples for possible applications as photoelectrochemical devices. However, some suggestions and corrections are required for publication in Royal Society Open Science, which are as follows:

- 1 - In the XRD results, inform the crystal structure of Bi₂S₃, based on JCPDS data.
- 2 - To enrich the work, add a figure outlining the sample preparation process.
- 3 - Inform in the preparation methodology that sample 1, 2 and 3 refer to heat treatments at 300, 400 and 500°C. This information is only in the legend of the XRD.
- 4 - If possible, add an image and composition by some technique, such as SEM / EDS.
- 5 - In figure 3, inform the gap of all samples. Associate the change in the sample gap with the results of photocurrent (figure 4) and impedance (figure 5).
- 6 - In figure 4 (b) indicate which sample the measure refers to. If possible, add the measurements of all analyzed samples.
- 7 - If possible, estimate the size of the bismuth nanocrystalline and the thickness of the Bi₂S₃ films.
- 8 - The figure legend can be further detailed with samples and measurement information. For example, the legend in figure 5 has no detailed information.

Decision letter (RSOS-200479.R0)

Dear Dr Zhao:

Title: Synthesis of Bi₂S₃ thin films based on pulse-plating bismuth nanocrystallines and its photoelectrochemical properties
Manuscript ID: RSOS-200479

The editor assigned to your manuscript has now received comments from reviewers. We would like you to revise your paper in accordance with the referee and Subject Editor suggestions which can be found below (not including confidential reports to the Editor). Please note this decision does not guarantee eventual acceptance.

Please submit your revised paper before 19-Jun-2020. Please note that the revision deadline will expire at 00.00am on this date. If we do not hear from you within this time then it will be assumed that the paper has been withdrawn. In exceptional circumstances, extensions may be possible if agreed with the Editorial Office in advance. We do not allow multiple rounds of revision so we urge you to make every effort to fully address all of the comments at this stage. If deemed necessary by the Editors, your manuscript will be sent back to one or more of the original reviewers for assessment. If the original reviewers are not available we may invite new reviewers.

On behalf of the Subject Editor Professor Anthony Stace and the Associate Editor Dr Darren Walsh.

RSC Associate Editor:
Comments to the Author:
(There are no comments.)

RSC Subject Editor:
Comments to the Author:
(There are no comments.)

Reviewers' Comments to Author:
Reviewer: 1

Comments to the Author(s)

This present work described a pulse-plating method for the synthesis of Bi₂S₃ thin films and also investigated the photoelectrochemical performance. In the present state, the manuscript is badly organized and there are still many problems to be addressed before acceptance for publication.

1. For the synthesis method, the authors claimed that the Bi₂S₃ thin films were prepared based on pulse-plating method in this paper for the first time. So, what is the remarkable progress compared to the electroplating method for the preparation of Bi₂S₃ thin films (J Solid State Electrochem 2009, 13, 1339-1350, 10.1007/s10008-008-0679-z)?

2. The Bi₂S₃ thin films on FTO surface were prepared primarily from the deposition of Bi films by using electroplating method, followed by thermal treatment for oxidation at different temperature and finally converted to Bi₂S₃ films. How can the bismuth oxides be guaranteed totally to be converted to Bi₂S₃ at 80 C for 10 min? From the evidence of PXRD patterns, the diffraction peaks of bismuth-metal films could be remarkably recorded being indexed to JCPDS: PDF number 44-1246. However, the characteristic peaks of Bi₂S₃ films as ascribed to JCPDS: PDF number 17-0320 were pretty weak, and they didn't provide other firm characterization for the chemical structure. How did the authors confirm the formation of Bi₂S₃ films deriving from the precursor oxides?

3. More experimental details should be provided. For examples, how were the UV-Vis adsorption spectra recorded? In transmission mode, reflection mode or other mode? What were the setup parameters for PXRD pattern performing? The testing conditions for photoelectrochemical measurements should be clearly described such as the purification of electrolyte solution by inert gas. The EIS tests should be clarified by providing the stable open-circuit potential value and frequency range.

4. The authors described that the reported method in the present work is able to deposit more uniform Bi₂S₃ thin films. However, from the SEM images, the roughness of Bi₂S₃ films seems very large. Could the authors compare the prepared Bi₂S₃ here to the previous works? Does the surface topography have the effect on the photoelectrochemical performance of Bi₂S₃ films?

5. This work showed the photoelectrochemical performance of as-synthesized Bi₂S₃ films. As demonstrated in literatures, the preferentially growing orientation of Bi₂S₃ nanorod along with [001] direction provides a rapid electron transport route for the photo-induced charge carriers (Dalton Trans., 2012, 41, 5581-5586; Crystengcomm. 2019, 21, 1474-1481). Do the authors could distinguish the orientation of Bi₂S₃ in the films? What do dominate the good photo-conversion efficiency of Bi₂S₃ films in the present work? How was the stability of Bi₂S₃ films, which is usually shown in the manuscript or electronic supplementary materials? How about the electron-hole separation efficiency compared to Bi₂S₃-based composite films with other semiconductors? The comparison of photoelectrochemical performance of as-prepared Bi₂S₃ films to the previous ones should also be discussed.

6. For the Nyquist plot, why did the linear part corresponding to the diffusion process keep in perpendicular to Z' axle rather than the line with a slop of around 1?

7. The language used in the manuscript should be largely polished as any grammar mistakes were found. Such as, "Nanostructured Bi₂S₃ thin film with high surface area and low light reflectivity is one of the most promising solar cells."

Reviewer: 2

Comments to the Author(s)

The authors report the synthesis and photoelectrochemical properties of Bi₂S₃ thin films samples for possible applications as photoelectrochemical devices. However, some suggestions and corrections are required for publication in Royal Society Open Science, which are as follows:

- 1 - In the XRD results, inform the crystal structure of Bi₂S₃, based on JCPDS data.
- 2 - To enrich the work, add a figure outlining the sample preparation process.
- 3 - Inform in the preparation methodology that sample 1, 2 and 3 refer to heat treatments at 300, 400 and 500oC. This information is only in the legend of the XRD.
- 4 - If possible, add an image and composition by some technique, such as SEM / EDS.
- 5 - In figure 3, inform the gap of all samples. Associate the change in the sample gap with the results of photocurrent (figure 4) and impedance (figure 5).
- 6 - In figure 4 (b) indicate which sample the measure refers to. If possible, add the measurements of all analyzed samples.
- 7 - If possible, estimate the size of the bismuth nanocrystalline and the thickness of the Bi₂S₃ films.
- 8 - The figure legend can be further detailed with samples and measurement information. For example, the legend in figure 5 has no detailed information.

Author's Response to Decision Letter for (RSOS-200479.R0)

See Appendix A.

RSOS-200479.R1 (Revision)

Review form: Reviewer 1

Is the manuscript scientifically sound in its present form?

Yes

Are the interpretations and conclusions justified by the results?

Yes

Is the language acceptable?

Yes

Do you have any ethical concerns with this paper?

No

Have you any concerns about statistical analyses in this paper?

No

Recommendation?

Accept as is

Comments to the Author(s)

All concerns have been addressed. The manuscript is acceptable for publication in present form.

Review form: Reviewer 2

Is the manuscript scientifically sound in its present form?

Yes

Are the interpretations and conclusions justified by the results?

Yes

Is the language acceptable?

Yes

Do you have any ethical concerns with this paper?

No

Have you any concerns about statistical analyses in this paper?

No

Recommendation?

Accept as is

Comments to the Author(s)

The authors performed the reviews and the manuscript is recommended for publication in RSOS.

Decision letter (RSOS-200479.R1)

Dear Dr Zhao:

Title: Synthesis of Bi₂S₃ thin films based on pulse-plating bismuth nanocrystallines and its photoelectrochemical properties
Manuscript ID: RSOS-200479.R1

It is a pleasure to accept your manuscript in its current form for publication in Royal Society Open Science. The chemistry content of Royal Society Open Science is published in collaboration with the Royal Society of Chemistry.

On behalf of the Subject Editor Professor Anthony Stace and the Associate Editor Dr Darren Walsh.

RSC Associate Editor:
Comments to the Author:
(There are no comments.)

RSC Subject Editor:
Comments to the Author:
(There are no comments.)

Reviewer(s)' Comments to Author:
Reviewer: 2

Comments to the Author(s)
The authors performed the reviews and the manuscript is recommended for publication in RSOS.

Reviewer: 1

Comments to the Author(s)
All concerns have been addressed. The manuscript is acceptable for publication in present form.

Appendix A

Original Manuscript ID: RSOS-200479

Original Article Title: “Synthesis of Bi₂S₃ thin films based on pulse-plating bismuth nanocrystallines and its photoelectrochemical properties”

To: Royal Society Open Science - Chemistry Editor

Re: Response to reviewers

Dear Editor:

Thank you for your letter and the reviewers' comments. Those comments are valuable and very helpful for revising and improving our paper. We have studied comments carefully and have made correction which we hope meet with approval. In the upload file, we edit the manuscript according (a) our point-by-point response to the comments (below) (response to reviewers) and (b) an updated manuscript with red highlighting indicating changes.

Best regards,

< Guochen Zhao > et al.

Reviewer#1, Concern#1: For the synthesis method, the authors claimed that the Bi₂S₃ thin films were prepared based on pulse-plating method in this paper for the first time. So, what is the remarkable progress compared to the electroplating method for the preparation of Bi₂S₃ thin films (J Solid State Electrochem 2009, 13, 1339–1350, 10.1007/s10008-008-0679-z)?

Our revision and response: Thank you very much for pointing this out. The simple pulse-plating method provides an efficient strategy to prepare the Bi₂S₃ thin films with regular nanostructures and large superficial area. Compared to the electroplating method for the preparation of Bi₂S₃ thin films, we don't use the toxic non-aqueous dimethyl sulfoxide medium, the complexing agent and the surfactant. Our process was carried out in aqueous electrolyte which is expected safer and more friendly to the environment, as well as more cost-effective. We have added this comparison according to the paper (J Solid State Electrochem 2009, 13, 1339–1350, 10.1007/s10008-008-0679-z). Revised portion are marked in red in the paper. Please see page 4, line 20-23. Moreover, we updated the manuscript by citing reference [26].

Reviewer#1, Concern#2: The Bi₂S₃ thin films on FTO surface were prepared primarily from the deposition of Bi films by using electroplating method, followed by thermal treatment for oxidation at different temperature and finally converted to Bi₂S₃ films. How can the bismuth oxides be guaranteed totally to be converted to Bi₂S₃ at 80 °C for 10 min? From the evidence of PXRD patterns, the diffraction peaks of bismuth-metal films could be remarkably recorded being indexed to JCPDS: PDF number 44-1246. However, the characteristic peaks of Bi₂S₃ films as ascribed to JCPDS: PDF number 17-0320 were pretty weak, and they didn't provide other firm characterization for the chemical structure. How did the authors confirm the formation of Bi₂S₃ films deriving from the precursor oxides?

Our revision and response: Thanks for your comment. The time of vulcanization treatment was investigated. In the range of 10 min to 1 h, there was no obvious change could be witnessed on XRD patterns. 10 min is the optimized processing time. As shown in figure (a) below, among the samples, there are no diffraction peaks of bismuth oxide (JCPDS: PDF number 74-1374). Since the thickness of Bi₂S₃ film is only around 300 nm, the diffraction is weak. The X-rays can easily penetrate and reach the interface of FTO substrate.

In order to figure out the composition of film, EDS analysis was carried out, which demonstrated the existence of elements Bi and S (the atomic ratio is around 2:3), see figure (b). The other element peaks belong to the FTO substrate. Thus, it can be conclude that the bismuth oxides are mostly converted to Bi₂S₃. Revised portion are marked in red in the paper. Please see page 7, line 3-12.

Reviewer#1, Concern#3: More experimental details should be provided. For examples, how were the UV-Vis adsorption spectra recorded? In transmission mode, reflection mode or other mode? What were the setup parameters for PXRD pattern performing? The testing conditions for photoelectrochemical measurements should be clearly described such as the purification of electrolyte solution by inert gas. The EIS tests should be clarified by providing the stable open-circuit potential value and frequency range.

Our revision and response: Thanks for your suggestion. We should provide more details. Revised portion are marked in red in the paper. Please see (page 5, line 36-41), (page 5, line 44-48) and (page 6, line 1-7).

Reviewer#1, Concern#4: The authors described that the reported method in the present work is able to deposit more uniform Bi₂S₃ thin films. However, from the SEM images, the roughness of Bi₂S₃ films seems very large. Could the authors compare the prepared Bi₂S₃ here to the previous works? Does the surface topography have the effect on the photoelectrochemical performance of Bi₂S₃ films?

Our revision and response: Your suggestion is very reasonable. The increase of surface roughness will enhance the reflection effect of incident light on the interface and thus improve the photo-conversion efficiency. We have added the comparison according to the suggestion. Revised portion are marked in red in the paper. Please see (page 9, line 16-18) and (page 12, line 7-15).

Reviewer#1, Concern#5: This work showed the photoelectrochemical performance of as-synthesized Bi₂S₃ films. As demonstrated in literatures, the preferentially growing orientation of Bi₂S₃ nanorod along with [001] direction provides a rapid electron transport route for the photo-induced charge carriers (Dalton Trans., 2012, 41, 5581-5586; Crystengcomm. 2019, 21, 1474-1481). Do the authors could distinguish the orientation of Bi₂S₃ in the films? What do dominate the good photo-conversion efficiency of Bi₂S₃ films in the present work? How was the stability of Bi₂S₃ films, which is usually shown in the manuscript or electronic supplementary materials? How about the electron-hole separation efficiency compared to Bi₂S₃-based composite films with other semiconductors? The comparison of photoelectrochemical performance of as-prepared Bi₂S₃ films to the previous ones should also be discussed.

Our revision and response: Thank you very much for your suggestion. In our work, the as-synthesized Bi₂S₃ belongs to the orthorhombic system, space group Pbnm. Intensities of (211), (221), (301) are relatively high. The optimization of plating parameter to control growing orientation will be our future work. The increase of surface roughness will enhance the reflection effect of incident light on the interface and thus improve the photo-conversion efficiency. The Nyquist results shows that the materials of the prepared Bi₂S₃ thin films as photo-anodes can significantly promote the effective separation of photo-generated electrons and holes comparison to Bi₂S₃-based composite films with other semiconductors. Revised portion are marked in red in the paper. Please see page12, line 12-15.

Reviewer#1, Concern#6: For the Nyquist plot, why did the linear part corresponding to the diffusion process keep in perpendicular to Z' axle rather than the line with a slop of around 1?

Our revision and response: Thanks. The slope coefficient of the straight line is relative to the ion transfer rate. Thus, the lower value of the charge transfer resistance

and the more vertical straight line are beneficial for enhancing the ion transport, and thus improving the electrochemical performance. Revised portions are marked in red in the paper. Please see page 13, line 1-6.

Reviewer#1, Concern#7: The language used in the manuscript should be largely polished as any grammar mistakes were found. Such as, “Nanostructured Bi_2S_3 thin film with high surface area and low light reflectivity is one of the most promising solar cells.”

Our revision and response: Thank you very much for your suggestion. We have revised the sentence to “Considering its high surface area and low light reflectivity, the prospect of nanostructured Bi_2S_3 thin film is promising for utilizing solar energy.” Please see page 3, line 11-13. Moreover, we updated the manuscript by improving the level of English.

Reviewer#2, Concern#1: In the XRD results, inform the crystal structure of Bi_2S_3 , based on JCPDS data.

Our revision and response: Thank you very much for your suggestion. The structure of Bi_2S_3 belongs to the orthorhombic system, space group Pbnm. We have revised it according to the suggestion. Revised portion are marked in red in the paper. Please see page 6, line 35.

Reviewer#2, Concern#2: To enrich the work, add a figure outlining the sample preparation process.

Our revision and response: Thank you very much for your suggestion. As shown in Figure 1, we have added a figure outlining the sample preparation process in the paper. Please see the Fig. 1.

Figure1. Schematic representation of the synthesis of Bi_2S_3 by pulse-plating method.

Reviewer#2, Concern#3: Inform in the preparation methodology that sample 1, 2 and 3 refer to heat treatments at 300, 400 and 500 °C. This information is only in the legend of the XRD.

Our revision and response: Thank you very much for your suggestion. We have revised it according to the suggestion. Revised portion are marked in red in the paper. Please see (page 8, line 38), (page 10, line 33-35), (page 11, line 37-39) and (page 13, line 34-36).

Reviewer#2, Concern#4: If possible, add an image and composition by some technique, such as SEM / EDS.

Our revision and response: Thank you very much for your good suggestion. As shown in figure 4, we have added the SEM image and the EDS pattern of Bi_2S_3 thin

film structures (sample 1 (300 °C)). Figure 4b indicated that only peaks of the elements Bi and S are present in the EDS spectrum, the other element peaks are belong to the FTO substrate. Revised portion are marked in red in the paper. Please see the Fig. 4 and page 9, line 1-3.

Figure 4. (a) SEM image and (b) EDS pattern of Bi_2S_3 thin film structures (sample 1 (300 °C))

Reviewer#2, Concern#5: In figure 3, inform the gap of all samples. Associate the change in the sample gap with the results of photocurrent (figure 4) and impedance (figure 5).

Our revision and response: Thank you very much for your suggestion. We have revised it according to the suggestion. Revised portion are marked in red in the paper. Please see page 9, line 44-53.

Reviewer#2, Concern#6: In figure 4 (b) indicate which sample the measure refers to. If possible, add the measurements of all analyzed samples.

Our revision and response: Thank you very much for your suggestion. We have revised it according to the suggestion. Please see the figure 6 (b).

Reviewer#2, Concern#7: If possible, estimate the size of the bismuth nanocrystalline and the thickness of the Bi_2S_3 films.

Our revision and response: Thank you very much for your suggestion. The thickness of the Bi_2S_3 film is about 297 nm according to the SEM image and the size of the bismuth nanocrystalline is about 25 nm according to the XRD data by Scherrer method, see the table below. Revised portion are marked in red in the paper. Please see page 6, line 14-19.

(hkl)	2 θ / deg	FWHM / deg	D / nm
(012)	27.170	0.225	35.920
(110)	39.697	0.348	24.000
(202)	48.897	0.506	17.054

Reviewer#2, Concern#8: The figure legend can be further detailed with samples and measurement information. For example, the legend in figure 5 has no detailed information.

Our revision and response: Thank you very much for your suggestion. We have revised all of the figure legend and given detailed information according to the suggestion. We updated the manuscript.